# Effect of Flumazenil on Emergence Agitation after Orthognathic Surgery: A Randomized Controlled Trial

**DOI:** 10.3390/jpm12030416

**Published:** 2022-03-07

**Authors:** Young Hyun Koo, Geun Joo Choi, Hyun Kang, Yong Hun Jung, Young Cheol Woo, Young-Jun Choi, Chong Wha Baek

**Affiliations:** 1Department of Anesthesiology and Pain Medicine, Chung-Ang University College of Medicine, 102 Heukseok-ro, Dongjak-gu, Seoul 06973, Korea; yhkoo28@caumc.or.kr (Y.H.K.); pistis23@cau.ac.kr (G.J.C.); roman00@cau.ac.kr (H.K.); jyh623@cau.ac.kr (Y.H.J.); wj12@cau.ac.kr (Y.C.W.); 2Department of Oral and Maxillofacial Surgery, Dental Center, Chung-Ang University College of Medicine, 102 Heukseok-ro, Dongjak-gu, Seoul 06973, Korea; omfs@cau.ac.kr

**Keywords:** emergence agitation, flumazenil, general anesthesia, orthognathic surgery, sevoflurane

## Abstract

Flumazenil, a gamma-aminobutyric acid receptor antagonist, can promote arousal even under general anesthesia without the use of benzodiazepines. We hypothesized that flumazenil could promote arousal and reduce emergence agitation in patients undergoing orthognathic surgery with sevoflurane anesthesia. One hundred and two patients were randomly allocated to the control or flumazenil group. Saline or flumazenil was administered at the end of the surgery. The incidence of emergence agitation was measured by using Aono’s four-point scale, with scores of 3 and 4 indicating emergence agitation. The primary outcome was the incidence of emergence agitation. Secondary outcomes included duration of emergence agitation and time intervals between the discontinuation of anesthetics, first response, extubation, and post-anesthesia care-unit discharge readiness. The incidence of emergence agitation was 58.3% and 38.9% in the control and flumazenil groups, respectively, but it was not statistically significant. However, the duration of emergence agitation was shorter in the flumazenil group (*p* = 0.012). There were no significant differences in the time intervals between the discontinuation of anesthetics, first response, and extubation. Although flumazenil did not reduce the incidence of emergence agitation in patients undergoing orthognathic surgery with sevoflurane anesthesia, it can be considered as an option for awakening patients in terms of improving emergence profiles.

## 1. Introduction

Emergence agitation is characterized by confusion, restlessness, and hyperactive physical behavior resulting from a temporary loss of consciousness during recovery from general anesthesia. It has an incidence of 0.25% to 90.5% and is considered a very important dilemma in postoperative care because it increases the risk of self-extubation, unintended removal of catheters, surgical site injury, and postoperative bleeding [1,2]. The risk of emergence agitation is high following orthognathic surgery, because the tracheal tube is maintained until the patient is fully awake to ensure airway security from facial or intra-oral edema, which can lead to a sense of suffocation [3,4]. 

In a previous study, when desflurane was used during orthognathic surgery, rapid awakening resulted in early situational awareness; therefore, emergence agitation was reduced compared to sevoflurane [5]. However, desflurane tends to increase the patient’s heart rate during anesthesia, and epinephrine injection during orthognathic surgery, administered to reduce the bleeding, can also cause tachycardia [6]. In contrast to desflurane, sevoflurane is not associated with heart-rate elevation during anesthesia [7]. Therefore, we attempted to determine a way to reduce emergence agitation while using sevoflurane. 

Meanwhile, flumazenil is a competitive inhibitor of γ-aminobutyric acid (GABA) receptors and reverses the hypnotic and sedative actions of benzodiazepines acting on GABA receptors. Recent studies have reported that arousal is promoted when flumazenil is administered, even when general anesthesia is administered through inhalation anesthetics or propofol and remifentanil [8,9,10,11]. Therefore, it can be hypothesized that flumazenil can aid in the recovery and reduce the incidence of emergence agitation in patients undergoing general anesthesia without using benzodiazepine. The purpose of this study was to investigate whether flumazenil can reduce emergence agitation in adult patients undergoing orthognathic surgery under general anesthesia, using sevoflurane. The primary outcome of this study was the incidence of emergence agitation. The secondary outcomes were the duration of emergence agitation and time intervals between the end of anesthesia, first response of arousal, extubation, and post-anesthetic care unit (PACU) discharge readiness.

## 2. Materials and Methods

### 2.1. Patients

This study was approved by the Institutional Review Board of Chung-Ang University Hospital, Seoul, Korea (registration number: 2130-008-456, date of approval: 17 May 2021) and was registered with Clinical Research Information Service (#KCT0006209). The American Society of Anesthesiologists physical status I or II patients aged 18–65 years who were scheduled for elective orthognathic surgery under general anesthesia were recruited. Patients weighing less than 45 kg or more than 100 kg, and those who took medications for neurological/psychological disorder, cardiopulmonary disease, or a history of these were excluded. Written informed consent was obtained from all participants. We fully explained the fact that all subjects may feel a sense of suffocation due to orofacial edema and the presence of a nasotracheal tube during arousal from anesthesia after surgery, and that severe discomfort may be experienced. Participants were blinded to their allocation group.

### 2.2. Randomization

According to a computer-generated table of random numbers, using randomization block size 6, participants were allocated to either the control (group C) or the flumazenil (group F) group randomly. Group allocations were sealed in sequentially numbered opaque envelopes by an investigator who was not involved in the perioperative management and outcome assessment. Another investigator who did not assess the outcome opened the envelope and prepared the experimental drug in a 5 cc syringe by the group allocation, which was either normal saline (3 mL) or flumazenil (0.3 mg, 3 mL).

### 2.3. Anesthetic Management

The standard monitoring methods, which included electrocardiography, non-invasive arterial pressure, pulse oximetry, and neuromuscular transmission, were applied after the patient entered the operating room. Propofol (2 mg/kg), fentanyl (2 µg/kg), and rocuronium (0.6 mg/kg) were administered intravenously to induce general anesthesia. Nasotracheal intubation was performed by using a 6.0 mm or 7.0 mm tracheal tube for women and men each. Then, with a steady fresh gas flow, sevoflurane with 50% N_2_O in oxygen was used to maintain anesthesia. The respiratory rate was regulated to maintain ETCO_2_ between 30 and 35 mmHg, and mechanical ventilation was controlled with a tidal volume of 6 mL/kg. Age-corrected minimal alveolar concentration (MAC) was monitored continuously by using a Carestation 650 A1 ventilator (GE Healthcare, Helsinki, Finland). Sevoflurane with oxygen/nitrous oxide in 0.5 FiO_2_ was administered at a 1.5–2.0 MAC to maintain anesthesia. Orthognathic surgery was performed by an experienced surgical team who was blinded to the group assignments. Before incision, a combination of 2% lidocaine and 1:200,000 epinephrine was administered to the oral submucosa. For bimaxillary orthognathic surgery, Le Fort I osteotomy was performed first, followed by short lingual osteotomy of the mandible. If tachycardia (>110 beats/min) and elevated blood pressure (systolic blood pressure > 120 mm/Hg) persisted even after increasing sevoflurane up to 2.0 MAC during the operation, additional fentanyl (25 µg) was administered intravenously twice, followed by labetaolol at 5 mg. Phenylephrine (50 mg) for hypotension (mean arterial pressure < 60 mmHg), esmolol 20 mg for tachycardia (>110 beats/min), and glycopyrrolate (0.2 mg) for bradycardia (<40 beats/min) were intravenously administered as required. 

Approximately 30 min before the end of surgery, fentanyl 1 µg/kg was intravenously administered, and sevoflurane was reduced to decrease the concentration to 1.0 MAC until the end of surgery. After confirmation that the train-of-four count was greater than 2, ventilation was controlled so that the patients could breathe spontaneously while maintaining an ETCO_2_ of 40 to 45 mmHg. Before the end of the operation, a syringe with the experimental drug was delivered to an anesthesia provider who was blinded to the group. At the completion of surgery, all anesthetics were discontinued with a fresh gas flow of 100% O_2_. Normal saline or flumazenil, sugammadex (2 mg/kg), and ramosetron (0.3 mg) were injected intravenously. During emergence, patients were not disturbed, except for gentle oral suction and repeated verbal requests to open their eyes. Once patients responded to verbal commands by opening their eyes or nodding their heads, and sufficient spontaneous breathing had been confirmed, extubation was performed. Subsequently, they were transferred to the PACU.

### 2.4. Outcome Assessments

An outcome assessor who was blinded to the patient group allocation continuously evaluated the emergence state, using a four-point categorical scale based on Aono’s scale: 1 = calm; 2 = not calm, but can be easily calmed by verbal instructions and able to tolerate ordinary fixation straps for both arms and legs; 3 = not calm despite frequent verbal instructions, moderately agitated or restless, and requires physical restraint; 4 = combative, excited, disoriented, and strongly requires physical restraint. Scores of 3 and 4 were regarded as emergence agitation [5,12]. Once emergence agitation occurred, the duration was recorded. The time interval from the end of anesthesia to the first response, including facial grimace, any movement of the head or extremities, and eye opening, was recorded and regarded as a deep plane of anesthesia. Similarly, the phase of the light plane was recorded as the time interval between the first response to extubation. The sum of the two phases (phases of the deep plane and light plane) was recorded as the time to extubation. If any adverse events occurred during the recovery phase from anesthesia, such as laryngospasm, desaturation, or self-extubation, these were also documented. 

In the PACU, an 11-point numeric scale, ranging from 0 (no pain) to 10 (worst pain imaginable), was used to evaluate postoperative pain. When a patient in the PACU complained of an NRS score of 5 or higher, 20 mg of nefopam was mixed with 100 mL of normal saline and administered intravenously. In the case of postoperative nausea or vomiting, 10 mg of metoclopramide was administered. The time when the modified Aldrete score exceeded 9 was recorded as PACU discharge readiness time, and all participants were questioned if they remembered their status during the recovery phase. 

### 2.5. Statistical Analysis

#### 2.5.1. Sample Size

The primary outcome measure was used to calculate sample size. In a previous study, the incidence of emergence agitation in the sevoflurane group was 70.8% [5]. Assuming that flumazenil decreased the incidence by 40% (absolute ratio of emergence agitation: 28.3%), and as a result of accepting α error as 5% and β error as 20%, the sample size was calculated to be 96, with 48 in each group. A dropout rate of 5% was applied, and 51 patients were included in each group.

#### 2.5.2. Data Analysis

Statistical analyses were conducted by using the SPSS ver. 23.0 (IBM Corp., Armonk, NY, USA). For intergroup comparisons, the distribution of data was first evaluated for normality, using the Shapiro–Wilk test. Normally distributed data were compared by using Student’s *t*-test, and non-normally distributed data were analyzed by using non-parametric tests. Descriptive variables were subjected to χ^2^ analysis or Fisher’s exact test, and statistical significance was set at *p* < 0.05. Data are presented as mean ± standard deviation (SD) or absolute number (%).

## 3. Results

Data were collected from May 2021 to December 2021. One hundred and six participants were assessed for eligibility, and after four exclusions, a total of 102 patients were randomized and completed the study (Figure 1). Table 1 summarizes the demographic features of patients and the duration of surgery in both groups. The incidence of emergence agitation, the primary outcome, seemed to be lower in the flumazenil group, but the difference was not significant (Figure 2). The emergence profiles are summarized in Table 2. Among the secondary outcomes, the duration of emergence agitation was significantly shorter in the flumazenil group (*p* = 0.012) (Figure 3). There were no significant differences in the time to first response, first response to extubation, and time to extubation. The time to PACU discharge readiness was significantly shorter in the flumazenil group (*p* = 0.04) (Figure 4). Pain scores and the requirement for rescue analgesics or anti-emetics at the PACU were comparable. There were no differences regarding hemodynamic changes between the number of patients who needed additional drugs, such as fentanyl, labetalol, esmolol, and phenylephrine. Self-extubation and intravenous line removal occurred in two patients in each of the control groups. 

## 4. Discussion

In this study, the incidence of emergence agitation showed a lower tendency when flumazenil was administered (58.3% vs. 38.9%), without reaching statistical significance. Instead, the duration of emergence agitation was shorter in the flumazenil group (2.5 ± 1.6 min) than in the control group (1.5 ± 0.8 min). 

There were no significant differences in the time to first response, extubation, and time interval between the first response and extubation, but they were slightly shorter in the flumazenil group. The time to PACU discharge was significantly shorter in the flumazenil group. This is thought to be caused by the antagonistic effect of flumazenil in patients under general anesthesia. In fact, several studies revealed that patients who were anesthetized with intravenous or inhalation anesthetics had faster recovery and reversal of amnesia when flumazenil was administered [8,9,10]. 

In Kim’s study, patients were anesthetized with sevoflurane/fentanyl, and 0.3 mg of flumazenil was administered in the same way as in this study. Spontaneous breathing, eye opening, and hand squeezing on verbal command, extubation, and date-of-birth recollection were significantly faster when flumazenil was administered [8]. The time to extubation in Kim’s study was 8.7 min in the control group and 8.0 min in the flumazenil group, whereas it was 8.2 and 9.7 min in our study, respectively. This difference occurred because the study protocols were different.

There are three possible mechanisms to explain the action of flumazenil in patients under general anesthesia without benzodiazepine administration. First, it is related to the GABA type A receptor, which is part of an important cellular mechanism involved in the unconsciousness, and anesthetics target this part of the central nervous system (CNS) [13]. It is not exactly known which subunit inhalation anesthetics, including sevoflurane and intravenous anesthetics, such as propofol, act on GABA type A receptors, and several studies have shown that they potentiate the activity of the GABA receptor [14,15]. As such, because anesthetics act on GABA receptors, flumazenil, an antagonist of the GABA receptor, can accelerate recovery from anesthesia. Second, flumazenil itself has an intrinsic direct CNS stimulant effect and can increase the bispectral index faster [9]. The last hypothesis is that flumazenil may act on endogenic benzodiazepine-like ligands (endozepines), which can be found even in patients who have not received benzodiazepines [16]. Endozepines have been found in various types of mammalian tissues, including cerebrospinal fluid, serum, plasma, urine, and breast milk [9,17,18,19,20].

Meanwhile, the characteristics and mechanisms of emergence agitation in adults are considered different from those in children. Emergence agitation in children is defined as a disturbance in a child’s awareness of attention to his or her environment, with disorientation and perceptual alteration, including hypersensitivity to stimuli and hyperactive motor behavior in the immediate post-anesthesia period, and it occurs frequently in the PACU [21,22]. Sevoflurane and desflurane, inhalation anesthetics commonly used in general anesthesia, have a low blood/gas distribution coefficient, allowing faster and more flexible control of anesthesia induction and awakening. Due to its rapid awakening, pain from surgery and anxiety in strange circumstances also appear quickly, so there have been reports of higher emergence agitation in children [23,24]. However, emergence agitation in adults only occurs in the acute phase in the operating room before attaining full consciousness after general anesthesia and ends when the patient can recognize his or her postoperative status [25]. 

Choi et al. concluded that, in contrast to children, adults with faster recovery of consciousness experience less agitation because they were given a detailed explanation of emergence prior to anesthesia and their rapid recognition of the post-operative environment. This might have led to a significantly lower incidence in the desflurane group than in the sevoflurane group [5]. In Choi et al.’s study, the time to extubation in the desflurane group was 3 min faster than that in the sevoflurane group (6 min vs. 9 min). In our study, although time to PACU discharge readiness was shorter in the flumazenil group, flumazenil did not reduce the time to extubation in the sevoflurane and desflurane groups in Choi et al.’s study. Therefore, the incidence of emergence agitation in this study was not significantly reduced. 

Emergence agitation is a hyperactive state that is related to the light plane of anesthetic depth. Therefore, we analyzed the time of arousal divided into time to first reaction and first reaction to extubation in order to differentiate the emergence phase into the phase of the light and deep planes. Both times were slightly shorter in the flumazenil group, but the difference was not significant. The time to PACU discharge readiness was reduced in the flumazenil group. It can be supposed that flumazenil accelerates awakening in patients throughout the overall phase, but the effect is weak and not sufficient to shorten each phase of arousal. In addition to not reducing the time to extubation sufficiently, flumazenil’s failure to shorten the phase of the light plane could be another reason for not reducing the incidence. 

Besides the reasons mentioned above, the reason why the incidence did not decrease is that flumazenil also has a partial benzodiazepine agonist-like effect. Several studies have shown that flumazenil has agonist activity against the GABA type A receptor and can potentiate the hypnotic activity of propofol [11,26,27,28]. Furthermore, one study reported that there was no antagonizing effect when flumazenil was administered to rats after anesthesia with halothane or propofol [26]. Therefore, it should be considered that, depending on the patient, contradictory results may be obtained.

However, when flumazenil was administered, the duration of emergence was significantly reduced. If emergence agitation is interpreted as a patient expressing discomfort in the light plane, the stimulus due to discomfort itself can promote awakening of the patient. In this dynamic situation, although the effect of flumazenil was insufficient to reduce the incidence or time to extubation, it is thought that the arousal effect of flumazenil was accentuated and caused a decrease in duration. 

There were some limitations to this study. Emergence agitation, according to Aono’s scale, refers to a state in which physical restraint is required and cannot be pacified with words, but scoring may vary depending on the evaluator, as it is a subjective evaluation. This may have led to the different agitation incidence of sevoflurane from the previous study. In a previous study by Choi et al., the incidence was 71%, whereas it was 58.3% in the control group in this study [5]. Changes in surgical technique and operation time may also have caused these differences. However, since each patient in the present study was evaluated by a single assessor, it is reasonable to analyze the differences between the two groups. Second, when measuring the duration of agitation and time to first response, extubation, and modified Aldrete score above 9, we only recorded the hour and minute of each event; hence, the statistics were based on minutes, not seconds. More thorough results may have been achieved if the data were recorded in seconds. Third, there may be issues with flumazenil dose and onset. The anesthetic agent was administered according to the body weight, but 0.3 mg of flumazenil was administered in batches. According to the inclusion criteria, the weight that can participate in the study was 45 to 100 kg; therefore, the weight difference between the study participants could be approximately two-fold. If the amount of flumazenil administered was altered according to body weight, the effect could be more pronounced. In addition, because the onset of flumazenil is approximately 1 to 2 min and the peak effect is 6 to 10 min, if flumazenil was injected according to the maximum effect time, the effect of flumazenil could be improved. In fact, when considering that the time to the modified Aldrete score above 9 was shorter in the flumazenil group, a more beneficial effect of flumazenil on the emergence profile can be expected if the administration timing is adjusted. Finally, the results of this study were confined to patients with orthognathic surgery who were anesthetized using sevoflurane and N_2_O. The effect of flumazenil on emergence agitation in various situations may differ depending on the anesthetic method and the type of surgery. Further studies are needed.

## 5. Conclusions

Although flumazenil did not reduce the incidence of emergence agitation in patients undergoing orthognathic surgery with sevoflurane anesthesia, it can be considered an option for awakening patients in terms of improving emergence profiles.

## Figures and Tables

**Figure 1 jpm-12-00416-f001:**
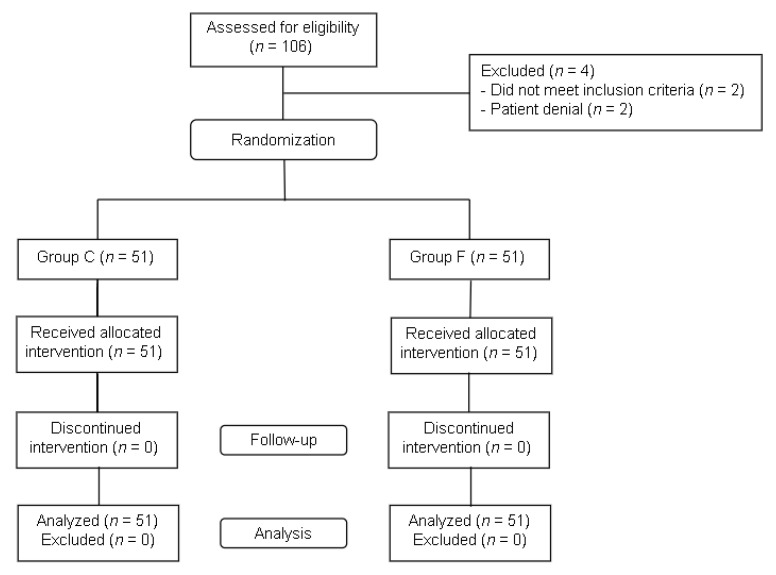
Flowchart of enrollment and randomization of the study.

**Figure 2 jpm-12-00416-f002:**
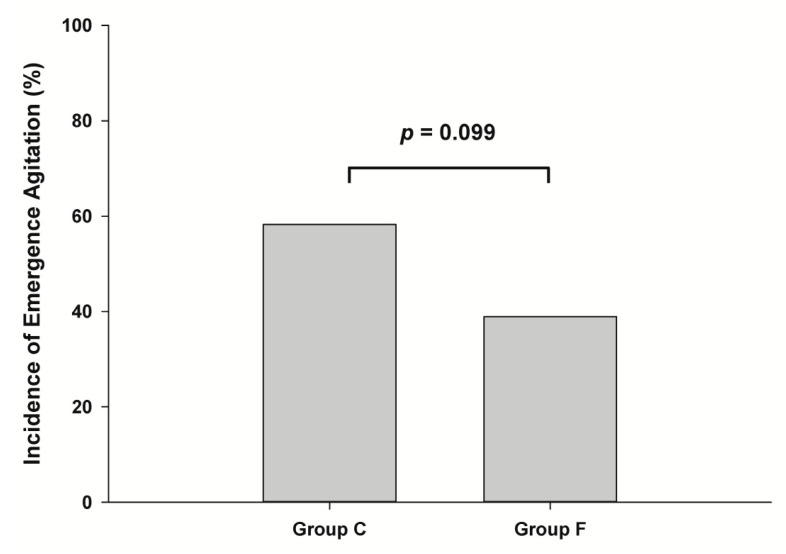
Bar graph showing incidence of emergence agitation (%). The incidence of emergence agitation was 58.3% and 38.9% in groups C and F, respectively. There were no significant differences (*p* = 0.099). Group C, control group; Group F, flumazenil group.

**Figure 3 jpm-12-00416-f003:**
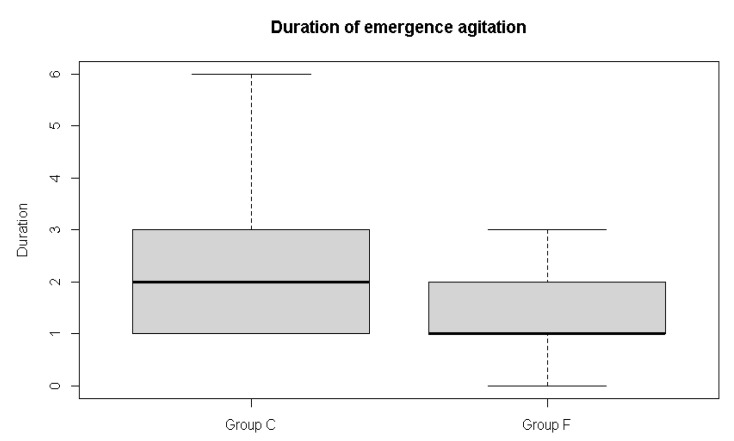
Box and whisker plot showing duration of emergence agitation in the control group and the flumazenil group. The median is represented as a line located in the middle of the box. The top and bottom of the box are the 75th and 25th percentiles, respectively, and the ends of the whiskers are the ±1.5 × interquartile range.

**Figure 4 jpm-12-00416-f004:**
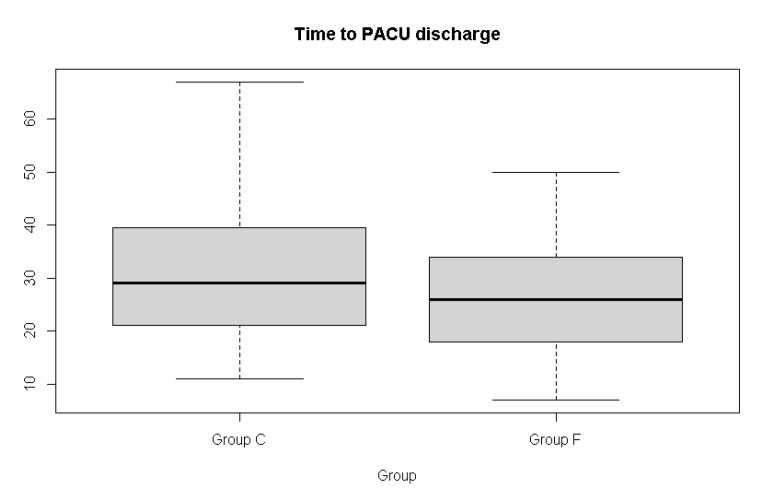
Box and whisker plot showing time to PACU discharge in the control group and the flumazenil group. The median is represented as a line located in the middle of the box. The top and bottom of the box are the 75th and 25th percentiles, respectively, and the ends of the whiskers are the ±1.5 × interquartile range.

**Table 1 jpm-12-00416-t001:** Demographic features of participants and duration of surgery.

	Group C (*n* = 51)	Group F (*n* = 51)	*p*-Value
Age (year)	25.3 ± 5.3	26.4 ± 6.9	0.436 †
Gender (M/F)	17/34	16/35	0.832
Height (cm)	167.0 ± 8.9	165.4 ± 7.3	0.316 †
Weight (Kg)	63.3 ± 13.4	60.33 ± 8.7	1.000
ASA Ⅰ/Ⅱ (*n*)	41/10	41/10	0.574 †
Duration of surgery (min)	184.1 ± 26.7	179.33 ± 26.2	0.290 †
Estimated blood loss (mL)	721.6 ± 83.9	717.7 ± 100.9	0.615 †

Data are presented as mean ± SD or absolute number (%). † Data were compared by using the Mann–Whitney U test because of abnormal distribution. Group C, control group; Group F, flumazenil group. There were no significant differences between the two groups.

**Table 2 jpm-12-00416-t002:** Characteristics of the emergence and recovery phases.

	Group C (*n* = 51)	Group F (*n* = 51)	*p*-Value
Incidence of emergence agitation	21 (58.3)	14 (38.9)	0.099
Duration of emergence agitation (min)	2.5 ± 1.6	1.5 ± 0.8	0.012 *†
Time to first response (min)	6.0 ± 3.9	5.18 ± 2.5	0.382 †
First response to extubation (min)	3.7 ± 2.6	3.0 ± 2.3	0.174 †
Time to extubation (min)	9.7 ± 4.8	8.2 ± 3.6	0.100 †
Time to PACU discharge readiness (min)	31.1 ± 12.7	26.4 ± 9.9	0.040 *†
NRS for pain in PACU	5.1 ± 1.6	5.0 ± 1.6	0.157 †
Analgesics in PACU	43 (84.3%)	41 (80.4%)	0.603
Anti-emetics in PACU	25 (49.0%)	27 (52.9%)	0.692

Data are presented as mean ± SD or absolute number (%). † Data were compared by using the Mann–Whitney U test because of abnormal distribution. Group C, control group; Group F, flumazenil group; NRS, numerical rating scale; PACU, post-anesthesia care unit. * *p* < 0.05, between-group comparisons.

## Data Availability

The datasets used and/or analyzed during the current study are available from the corresponding author upon reasonable request.

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
