# Peer review of "Effect of Flumazenil on Emergence Agitation after Orthognathic Surgery: A Randomized Controlled Trial"

_jpm, 2022, doi:10.3390/jpm12030416_

Round 1

Reviewer 1 Report

This RCT aims to compare the incidence of emergence agitation after othognatic surgery under sevoflurane anaesthesia with/ without consecutive administration of flumazenil. Secondary goal is the comparison of quantitative parameters (time periods) during recovery from anaesthesia.

The present study focusses on clinical aspects and potential benefits of flumazenil administration after sevoflurane anaesthesia. This approach seems to be conclusive, as there is some evidence on this topic, which the authors have correctly pointed out. The particular benefits of reducing the incidence of emergence agitation in context of orthognatic surgery are outlined clearly in the manuscript. The authors' conclusions are appropriate to the design, the findings, and the discussion in the present study.

Specific comments:

L. 158-160: according to the statistical analysis, there is no significant difference; thus the manuscript should not suggest that the incidence in flumazenil group was lower; perhaps the wording "seemed to be lower" might be more appropriate;

L.295-299: this passage appears to be difficult to understand; I can't see a clear statement/ message here; consider editing the wording or avoiding this part of the text (L. 297 - it has to read "significantly");

Reviewer 2 Report

On P3L107 - Not clear why the study syringes need to be handed to the outcome assessor - appears like the study drug is given at the same time as reversal and antiemetic administration by the anesthesia provider - If such is the design of the study, then the outcome assessor's role need just be to assess the recovery profile, not handle or administer the drug. Ideally the assessor should only enter after the study drug administration.

P3L140: Not sure what is intended by the statement that the primary measure outcome is sample size. Instead please rephrase that the intent is to estimate sample size or power calculation
